# Conformally Invariant Gravity and Gravitating Mirages

**Victor Berezin [†] and Inna Ivanova \*,[†]**

Institute for Nuclear Research of the Russian Academy of Sciences, 60th October Anniversary Prospect 7a, 117312 Moscow, Russia; berezin@inr.ac.ru

\* Correspondence: pc_mouse@mail.ru

[†] These authors contributed equally to this work.

**Abstract:** The action of an ideal fluid in Euler variables with a variable number of particles is used for the phenomenological description of the processes of particle creation in strong external fields. It has been demonstrated that the conformal invariance of the creation law imposes quite strict restrictions on the possible types of sources. It is shown that combinations with the particle number density in the creation law can be interpreted as dark matter within the framework of this model.

**Keywords:** conformal invariance; perfect fluid; dark matter; cosmology

## 1. Introduction

Conformal invariance is a good candidate for the role of a fundamental symmetry, which, along with other symmetries, increases the likelihood of the Universe emerging from "nothing" [1]. Similar ideas are supported by many researchers such as Roger Penrose [2] and Gerard 't Hooft [3].

The conformally invariant gravitational Lagrangian contains terms that are quadratic in curvature. The results found by several independent research groups [4–9] show that such terms are linked to the conformal anomaly responsible for the particle creation. The conformal anomaly can be included in the action integral, where it consists of two parts: local and nonlocal. The local part is included in the gravitational Lagrangian as a set of counterterms and in the one-loop approximation is equal to the sum of the quadratic terms in the Riemann curvature tensor and its convolutions.

The study of particle production processes in the presence of strong external fields plays an important role both in cosmology and in black hole physics. It is especially difficult to calculate the back reaction for these problems, because it is necessary to take into account not only the influence on the metric from already produced particles, but also from vacuum polarization.

The exact solution of the quantum problem requires boundary conditions, and the latter can be imposed only after solving field equations with the energy–momentum tensor obtained by appropriate averaging from the quantum problem. In order to avoid these obstacles, we consider a phenomenological description of particle creation processes. It is a quantum process, but classical description is possible when the external fields are strong enough and the separation between just-created particles becomes of the order of their Compton length, and we can safely approximate them with some condensed matter. For example, F. Hoyle [10] used a classical creation field in order to introduce the idea of the continuous creation of matter. The thermodynamic approach to particle production at the expense of a gravitational field has been studied in [11]. Recently, J. Farnes [12] applied Hoyle's creation tensor and the concept of negative mass to propose a single negative-mass fluid explanation of dark matter and dark energy.

In the phenomenological approach to particle creation, the nonlocal processes become, formally, the local ones. The same concerns also the trace anomalies, and, for example, in the article [13], it is shown that the non-local terms in the effective action become insignificant

under certain conditions. In this case, the use of anomaly-induced effective action can be considered as an example of a phenomenological description of particle production.

To describe the processes of particle creation in the presence of strong external fields, we use the action for an ideal fluid in Euler variables [14], in which the particle conservation law is replaced by the creation law [15]. This method makes it possible to study the process of particle creation phenomenologically at the classical level but while also taking into account the back reaction. In addition, it will be shown further that the use of the conformally invariant action of gravity in combination with the considered action of matter leads to a case in which we are actually dealing with a kind of Sakharov's induced gravity [16].

When applying the model in consideration of cosmology, it can be assumed that it is most relevant for those phases of the evolution of the Universe when there was a rapid birth of particles. Take, for example, immediately after the supposed birth of the Universe, "nothing" [1], or at the end of inflation during the reheating [17]. Moreover, if the Universe was born anisotropic [18], then, as shown in the articles [19–21], it was the birth of particles that led to its isotropization.

It is easy to verify that the law of particle creation is itself conformally invariant. If we assume that the source of particle creation is an external scalar field, then we obtain fairly strict restrictions on the possible types of sources. Specifically, they include conformally invariant combinations of geometric quantities, scalar fields, and particle number density. It turns out that it is the combinations with the particle number density that contribute to the hydrodynamic part of the energy-momentum tensor and act like dust and radiation. It is important to note that the above types of sources are not real matter but rather an echo of the quantum process of particle creation. In this regard, their interpretation as dark matter becomes possible.

## 2. Local Conformal Transformation

This paper considers Riemannian geometry, which is completely determined by specifying the metric $g_{\mu\nu}$. The affine connection $\Gamma^{\lambda}_{\mu\nu}(x)$ is specified using Christoffel symbols:

$$\Gamma^{\sigma}_{\mu\nu} = \Gamma^{\sigma}_{\nu\mu}, \quad g_{\mu\nu;\sigma} = 0, \quad \Gamma^{\sigma}_{\mu\nu} = \frac{1}{2} g^{\sigma\lambda} \left( g_{\mu\lambda,\nu} + g_{\nu\lambda,\mu} - g_{\mu\nu,\lambda} \right), \tag{1}$$

It defines the parallel transport of vectors and tensors and their covariant derivatives

$$l^{\mu}_{;\lambda} = l^{\mu}_{,\lambda} + \Gamma^{\mu}_{\lambda\nu} l^{\nu}, \tag{2}$$

where "comma" denotes a partial derivative while "semicolon" denotes covariant derivative.

The Riemann tensor $R^{\mu}{}_{\nu\lambda\sigma}$ is defined as follows:

$$R^{\mu}{}_{\nu\lambda\sigma} = \frac{\partial \Gamma^{\mu}_{\nu\sigma}}{\partial x^{\lambda}} - \frac{\partial \Gamma^{\mu}_{\nu\lambda}}{\partial x^{\sigma}} + \Gamma^{\mu}_{\varkappa\lambda} \Gamma^{\varkappa}_{\nu\sigma} - \Gamma^{\mu}_{\varkappa\sigma} \Gamma^{\varkappa}_{\nu\lambda}, \tag{3}$$

Ricci tensor $R_{\mu\nu}$ is its convolution:

$$R_{\mu\nu} = R^{\lambda}{}_{\mu\lambda\nu}. \tag{4}$$

The curvature scalar is $R = g^{\mu\nu} R_{\mu\nu}$.

Next let us consider a local conformal transformation; by definition we have:

$$ds^2 = \Omega^2(x) d\hat{s}^2 = \Omega^2(x) \hat{g}_{\mu\nu} dx^{\mu} dx^{\nu}, \tag{5}$$

where $\Omega(x)$ is the conformal factor, and "hats" denotes the conformally transformed quantities.

It is worth noting that a local conformal transformation is fundamentally different from a coordinate change. Different coordinates correspond to different observers, but the geometry of space-time itself remains unchanged, in contrast to a local conformal transformation, which does not change the coordinates, but changes the geometry.

The metric and its determinant are transformed, evidently, in the following way:

$$g_{\mu\nu} = \Omega^2 \widehat{g}_{\mu\nu}, \quad g^{\mu\nu} = \frac{1}{\Omega^2} \widehat{g}^{\mu\nu}, \quad \sqrt{-g} = \Omega^4 \sqrt{-\widehat{g}}. \tag{6}$$

An important geometric quantity that will be used below is the Weyl tensor $C_{\mu\nu\lambda\sigma}$, which is the traceless part of the Riemann tensor,

$$C_{\mu\nu\lambda\sigma} = R_{\mu\nu\lambda\sigma} - \frac{1}{2}R_{\mu\lambda}\,g_{\nu\sigma} + \frac{1}{2}R_{\mu\sigma}\,g_{\nu\lambda} - \frac{1}{2}R_{\nu\sigma}\,g_{\mu\lambda} + \frac{1}{2}R_{\nu\lambda}\,g_{\mu\sigma} + \frac{1}{6}R\big(g_{\mu\lambda}\,g_{\nu\sigma} - g_{\mu\sigma}\,g_{\lambda\nu}\big).$$

In the context of this work, its most important property is conformal invariance:

$$C^{\mu}{}_{\nu\lambda\sigma} = \hat{C}^{\mu}{}_{\nu\lambda\sigma}. \tag{7}$$

## 3. Phenomenological Description of Particle Creation

There are two types of dynamical variables in classical hydrodynamics: Lagrangian and Eulerian. The first ones are tied to the motion of individual particles, so the world line of each particle is subject to variation when applying the principle of least action. These coordinates are not suitable for describing the processes of creation or annihilation, and therefore, the Euler formalism is preferred, when dynamical variables are fields describing the average characteristics of the medium. This formalism was developed by J. R. Ray [14], who showed that the motion equation for an ideal fluid derived from this action coincides with the Euler equation. The advantage of this approach is that the continuity equation is explicitly incorporated into the action through the corresponding connection with the Lagrange multiplier.

Let us consider the action of an ideal fluid in Euler variables [14],

$$\begin{aligned} S_{\mathrm{m}} \;=\; & -\int \varepsilon(X, n)\sqrt{-g}\,d^4x + \int \lambda_0 (u_\mu u^\mu - 1)\sqrt{-g}\,d^4x + \int \lambda_1 (nu^\mu)_{;\mu}\sqrt{-g}\,d^4x + \\ & + \int \lambda_2 X_{,\mu} u^\mu \sqrt{-g}\,d^4x. \end{aligned} \tag{8}$$

The dynamical variables are the particle number density $n(x)$, the four-velocity $u^\mu(x)$, and the auxiliary dynamical variable $X(x)$ introduced in order to avoid the identically zero vorticity of particle flow. From the constraint with the Lagrangian multiplier $\lambda_2$, it follows that $X(x)$ is constant along the trajectories, and therefore, the choice of this function defines the labeling of the trajectories.

The energy density $\varepsilon$ provides us with the equation of state $p = p(\varepsilon)$, where

$$p = n\frac{\partial \varepsilon}{\partial n} - \varepsilon, \tag{9}$$

is the hydrodynamic pressure.

The corresponding constraints are obtained by varying the matter action with respect to the Lagrangian multipliers $\lambda_0$, $\lambda_1$, and $\lambda_2$, the four velocity normalization $u^\mu u_\mu = 1$, the particle number conservation $(nu^\mu)_{;\mu} = 0$ and the enumeration of trajectories $X_{,\mu} u^\mu = 0$, respectively.

The energy momentum tensor is:

$$T^{\mu\nu} = (\varepsilon + p)u^\mu u^\nu - pg^{\mu\nu}. \tag{10}$$

As demonstrated in the article by [15], the process of particle creation can be described phenomenologically if the corresponding constraint in the action of an ideal fluid is modified:

$$(nu^\mu)_{;\mu} = \Phi(\text{inv}), \tag{11}$$

where function $\Phi$ depends on the invariants of the fields responsible for the creation process.

It is easy to show that the left-hand side of the creation law becomes conformally invariant when multiplied by the root of the modulus of the determinant of the metric:

$$n = \frac{\hat{n}}{\Omega^3}, \quad u^\mu = \frac{\hat{u}^\mu}{\Omega}, \quad \sqrt{-g} = \Omega^4 \sqrt{-\hat{g}}, \tag{12}$$

hence

$$\begin{aligned}
(nu^\mu)_{;\mu} &= \frac{1}{\sqrt{-g}}(nu^\mu \sqrt{-g})_{,\mu} = \frac{1}{\sqrt{-g}}\left(\frac{\hat{n}}{\Omega^3}\frac{\hat{u}^\mu}{\Omega}\Omega^4 \sqrt{-\hat{g}}\right)_{,\mu} = \\
&= \frac{1}{\sqrt{-g}}(\hat{n}\hat{u}^\mu \sqrt{-\hat{g}})_{,\mu},
\end{aligned} \tag{13}$$

Then, it follows that, in turn, the quantity $\Phi\sqrt{-g}$ is also conformally invariant.

In the absence of classical external fields, the birth of particles is due to the vacuum polarization caused by gravity, so $\Phi$ is a function of geometric invariants. In Riemannian geometry in the four-dimensional case, the square of the Weyl tensor $C^2 = C_{\mu\nu\lambda\sigma} C^{\mu\nu\lambda\sigma}$ is the only possible choice if we restrict ourselves to invariants which are at most quadratic in the curvature tensor. The same result was obtained in [22] for particle creation by the vacuum fluctuations of the massless scalar field on the background of the homogeneous and slightly anisotropic cosmological spacetime. For our model, it is universal for any Riemannian geometry, irrespective of the form of the gravitational Lagrangian, and the back reaction is also taken into account. In this regard, it can be assumed that the creation law describes the relationship between the vacuum average values of the corresponding quantities.

If we consider a case in which some external scalar field $\varphi$ is involved in the creation process, then additional possible contributions to the source function $\Phi$ appear:

$$\varphi\Box\varphi - \frac{1}{6}\varphi^2 R + \Lambda\varphi^4, \tag{14}$$

It is easy to see that it is invariant under a conformal transformation when the scalar field changes as

$$\varphi = \frac{\hat{\varphi}}{\Omega}, \tag{15}$$

where $\Box$ denotes Laplace-–Beltrami operator.

The particles in question are on shell quanta of the scalar field, so they can also produce "new" particles. The rate of particle creation in this case should depend on the density of the number of "old" particles, i.e., it is some function of n. Due to conformal invariance, the most natural choice is $\varphi\,n$, and $n^{\frac{4}{3}}$. It is easy to verify that, when multiplied by $\sqrt{-g}$, they form conformal invariants. Theoretically, it is possible to use other degrees of $\varphi$ and $n$, but this leads to the appearance of particles with the properties of exotic or phantom matter, so we will limit ourselves to the options presented above. Thus, our creation law takes the following form:

$$\Phi = \alpha\,C^2 + \beta\left(\varphi\Box\varphi - \frac{1}{6}\varphi^2 R + \Lambda\varphi^4\right) + \gamma_1\,\varphi\,n + \gamma_2\,n^{\frac{4}{3}}. \tag{16}$$

## 4. Induced Gravity

Let us consider the action of an ideal fluid modified in the manner indicated earlier:

$$\begin{aligned}
S_{\mathrm{m}} = &-\int \varepsilon(X, \varphi, n)\sqrt{-g}\,d^4x + \int \lambda_0(u_\mu u^\mu - 1)\sqrt{-g}\,d^4x + \\
&+ \int \lambda_1\big((nu^\mu)_{;\mu} - \Phi\big)\sqrt{-g}\,d^4x + \int \lambda_2 X_{,\mu} u^\mu \sqrt{-g}\,d^4x,
\end{aligned} \tag{17}$$

note that $\varepsilon = \varepsilon(X, \varphi, n)$.

Let us consider a situation in which the action of gravity is conformally invariant. This case is to a certain extent equivalent to induced gravity, in which there is nothing except the action of matter, since the Lagrangian multiplier $\lambda_1$ is defined up to a constant; therefore, even in the absence of a separate Lagrangian for gravity, we can distinguish terms proportional to $C^2$ and $\varphi^2 R$. For the first time, such models, in which there is no separate action for gravity, were studied by A.D. Sakharov [16]. He suggested that the gravitational field is not fundamental, but is the result of the averaged influence of the vacuum fluctuations of all other quantum fields; these ideas formed the basis of the theory of induced gravity. Thus, we assume:

$$S_m = S_{tot}. \tag{18}$$

Evidently,

$$\frac{\delta S_{tot}}{\delta \Omega} = \frac{\delta S_m}{\delta \Omega} = 0, \tag{19}$$

in the solutions. The only part of the action of matter that is not conformally invariant from the very beginning or does not vanish due to constraints is

$$\int \varepsilon(X, \varphi, n) \sqrt{-g}\, d^4x. \tag{20}$$

Since $n = \frac{\hat{n}}{\Omega^3}$, $\quad \varphi = \frac{\hat{\varphi}}{\Omega}$, $\quad \sqrt{-g} = \Omega^4 \sqrt{-\hat{g}}$, one gets:

$$\varphi\, \frac{\partial \varepsilon}{\partial \varphi} + 3n\, \frac{\partial \varepsilon}{\partial n} = 4\, \varepsilon, \tag{21}$$

with the solution:

$$\varepsilon = F\left(\frac{n}{\varphi^3}\right) \varphi^4, \tag{22}$$

where F is an arbitrary function of one variable.

There are two important examples. For dust, that is, for $p = 0$, it follows from this equation that $\varepsilon = \mu_0\, n\, \varphi$, where $\mu_0$ is a constant. For radiation, $\varepsilon = 3p$, therefore, two options are possible: either $\varphi = 0$, or $\frac{\partial \varepsilon}{\partial \varphi} = 0$. That is, the energy density does not depend on the scalar field: $\varepsilon = \nu_0\, n^{\frac{4}{3}}$. Note the resemblance with two "hydrodynamical" terms in the creation law.

## 5. Equations of Motion and Constraints

Let us derive the (modified) hydrodynamical equations of motion and constraints for the action in question:

$$S_{\mathrm{m}} = -\int \varepsilon(X, \varphi, n) \sqrt{-g}\, d^4x + \int \lambda_0 (u_\mu u^\mu - 1) \sqrt{-g}\, d^4x + \int \lambda_2 X_{,\mu} u^\mu \sqrt{-g}\, d^4x +$$

$$+ \int \lambda_1 \left( (nu^\mu)_{;\mu} - \gamma_1\, \varphi\, n - \gamma_2\, n^{\frac{4}{3}} - \alpha\, C^2 - \beta\left( \varphi \Box \varphi - \frac{1}{6}\, \varphi^2\, R + \Lambda\, \varphi^4 \right) \right) \sqrt{-g}\, d^4x.$$

Dynamical variables are n, $u^\mu$, $\varphi$, and X:

$$\delta\varphi: \quad \beta\left( \lambda_1 \Box \varphi + \Box(\lambda_1 \varphi) + 4\lambda_1\, \Lambda \varphi^3 - \frac{1}{3}\lambda_1\, \varphi\, R \right) + \gamma_1\, n = -\frac{\partial \varepsilon}{\partial \varphi}, \tag{23}$$

$$\delta n: \quad -\frac{\partial \varepsilon}{\partial n} - \lambda_{1,\sigma}\, u^\sigma - \lambda_1 \gamma_1\, \varphi - \frac{4}{3}\lambda_1 \gamma_2\, n^{\frac{1}{3}} = 0, \tag{24}$$

$$\delta u^\mu: \quad \lambda_2\, X_{,\mu} + 2\lambda_0\, u_\mu - \lambda_{1,\mu}\, n = 0, \tag{25}$$

$$\delta X: \quad -\frac{\partial \varepsilon}{\partial X} - (\lambda_2\, u^\sigma)_{;\sigma} = 0. \tag{26}$$

The corresponding constraints are:

$$\delta\lambda_0: \quad u_\sigma u^\sigma - 1 = 0, \tag{27}$$

$$\delta\lambda_1: \quad (nu^\sigma)_{;\sigma} = \Phi, \tag{28}$$

$$\delta\lambda_2: \quad X_{,\sigma} u^\sigma = 0. \tag{29}$$

From Equation (25) multiplied by $u^\mu$ and constraints we get:

$$2\lambda_0 = -n\frac{\partial\varepsilon}{\partial n} - \lambda_1\gamma_1\,\varphi\,n - \frac{4}{3}\lambda_1\gamma_2\,n^{\frac{4}{3}}. \tag{30}$$

Let us calculate the hydrodynamical part of the energy–momentum tensor, that is, the energy–momentum tensor of the perfect fluid plus contribution from the $\gamma_1$ and $\gamma_2$ terms. From the general definition:

$$S_m = -\frac{1}{2}\int T^{\mu\nu}\,\delta g_{\mu\nu}\,\sqrt{-g}\,d^4x, \tag{31}$$

Taking into account Equation (30), we get:

$$T^{\mu\nu}_{hydro} = \varepsilon\,g^{\mu\nu} - 2\lambda_0\,u^\mu u^\nu + g^{\mu\nu}\left(n\,\lambda_{1,\sigma}\,u^\sigma + \lambda_1\gamma_1\,\varphi\,n + \lambda_1\gamma_2\,n^{\frac{4}{3}}\right) =$$
$$= \left(\varepsilon + p + \lambda_1\gamma_1\,\varphi\,n + \frac{4}{3}\,\lambda_1\gamma_2\,n^{\frac{4}{3}}\right)u^\mu u^\nu - g^{\mu\nu}\left(p + \frac{1}{3}\,\lambda_1\gamma_2\,n^{\frac{4}{3}}\right). \tag{32}$$

The remaining parts of the energy–momentum tensor are:

$$T^{\mu\nu}[\varphi] = \lambda_1\beta\Lambda\,\varphi^4\,g^{\mu\nu} - \beta\,\partial_\sigma(\lambda_1\varphi)\partial^\sigma\varphi\,g^{\mu\nu} + \beta\,\partial^\mu(\lambda_1\varphi)\partial^\nu\varphi +$$
$$+ \beta\,\partial^\nu(\lambda_1\varphi)\partial^\mu\varphi + \frac{\beta}{3}\left\{\lambda_1\varphi^2\,G^{\mu\nu} - \nabla^\mu\nabla^\nu\left(\lambda_1\varphi^2\right) + g^{\mu\nu}\,\Box\left(\lambda_1\varphi^2\right)\right\},$$
$$T^{\mu\nu}[C^2] = -8\alpha\left(\nabla_\sigma\nabla_\eta + \frac{1}{2}\,R_{\sigma\eta}\right)(\lambda_1\,C^{\mu\sigma\nu\eta}), \tag{33}$$

where $G^{\mu\nu}$ is the Einstein tensor. Since we are dealing with induced gravity, then:

$$T^{\mu\nu} = T^{\mu\nu}_{hydro} + T^{\mu\nu}[\varphi] + T^{\mu\nu}[C^2] = 0. \tag{34}$$

It should be clarified that the trace of the energy–momentum tensor $T$ is equal to zero, even for a non-zero gravitational part of the action, if it is conformally invariant. Let us show that the condition $T = 0$ reduces to Equation (21) obtained above:

$$T = \varepsilon - 3p + 4\beta\,\lambda_1\,\Lambda\,\varphi^4 - \frac{\beta}{3}\,\lambda_1\,\varphi^2\,R + \beta\,\varphi\,\Box(\lambda_1\varphi) + \beta\,\lambda_1\varphi\,\Box\varphi + \lambda_1\,\gamma_1\,\varphi n =$$
$$= \varepsilon - 3p - \varphi\frac{\partial\varepsilon}{\partial\varphi} = 4\varepsilon - 3n\frac{\partial\varepsilon}{\partial n} - \varphi\frac{\partial\varepsilon}{\partial\varphi}, \tag{35}$$

where in the second equality, the equation of motion obtained by variation in $\varphi$ was used.

The terms from the creation law which contain the particle number density lead to the appearance of corresponding contributions to the hydrodynamic part of the energy–momentum tensor: the term with $\gamma_1$ is dust-like and the term with $\gamma_2$ is radiation-like. They are not real because the particle number density n refers to real created particles whose equation of state can be arbitrary (anything). We can say that they are echoes of the process of creation itself. Thus, the most appropriate name for them is "gravitating mirages".

Finding a general solution to the equations of motion is quite a difficult task, so we will limit ourselves to considering two special cases: $\varphi = 0$ and $\lambda_1 = const$.

In the first case, Equation (21) implies that our perfect fluid is radiation, then, according to (23), either $n$ or $\gamma_1$ is zero. If $\gamma_1 = 0$, then it follows from (24) and (32) that:

$$\lambda_{1,\sigma}\, u^\sigma = -\frac{4}{3}\, n^{\frac{1}{3}}\, (\nu_0 + \lambda_1\, \gamma_2), \tag{36}$$

$$T_{hydro}^{\mu\nu} = \frac{1}{3}\, n^{\frac{4}{3}}\, (\nu_0 + \lambda_1\, \gamma_2)\, (4\, u^\mu u^\nu - g^{\mu\nu}). \tag{37}$$

Using the gauge $n = n_0 = const$ and the comoving coordinate system, where $u^\sigma = \delta_0^\sigma$, we can find $\lambda_1$ considering that it depends only on the proper time $t$:

$$\lambda_1(t) = -\frac{\nu_0}{\gamma_2} + \left(\lambda_1(0) + \frac{\nu_0}{\gamma_2}\right) exp\left\{-\frac{4}{3}\, \gamma_2\, n_0^{\frac{1}{3}}\, t\right\}. \tag{38}$$

Note that $\lambda_1$ tends to a constant $-\frac{\nu_0}{\gamma_2}$, while $t \to \infty$ if $\gamma_2 > 0$.

In the second case from Equation (24), we get:

$$\frac{\partial \varepsilon}{\partial n} = -\lambda_1 \left(\gamma_1\, \varphi + \gamma_2\, n^{\frac{1}{3}}\right), \tag{39}$$

the solution is:

$$\varepsilon = -\lambda_1 \left(\gamma_1\, n\varphi + \gamma_2\, n^{\frac{4}{3}}\right) + f(\varphi). \tag{40}$$

Function $f(\varphi)$, then, can be found from the relation (22):

$$f(\varphi) = C\varphi^4, \tag{41}$$

where C is an arbitrary constant. The hydrodynamical part of the energy–momentum tensor is: $T_{hydro}^{\mu\nu} = C\, \varphi^4\, g^{\mu\nu}$. This means that in this case, the term $f(\varphi)$ in $\varepsilon$ is equivalent to the shift of the constant $\Lambda$. The equation of motion for $\varphi$ reduces to the following:

$$2\lambda_1\beta\left(\Box\varphi - \frac{1}{6}R\,\varphi + 2\Lambda\,\varphi^3\right) + 4C\varphi^3 = 0. \tag{42}$$

The conformal invariance of the equations of motion and the creation law makes it possible to simplify the problem by fixing the gauge. In the gauge $\varphi = \varphi_0 = const$ from the Equation (42) we get:

$$R = \frac{12\varphi_0^2}{\beta\lambda_1}\, (C + \lambda_1\, \beta\Lambda) = const, \tag{43}$$

therefore the space-time in question is equivalent to the geometry with constant scalar curvature up to a conformal factor.

## 6. Cosmology

Let us consider cosmological solutions by which we understand the homogeneous and isotropic space-times described by the Robertson–Walker metric:

$$ds^2 = dt^2 - a^2(t)dl^2, \tag{44}$$

$$dl^2 = \gamma_{ij}dx^i dx^j = \frac{dr^2}{1 - kr^2} + r^2(d\theta^2 + \sin^2\theta d\varphi^2), \quad (k = 0, \pm 1),$$

with the scale factor $a(t)$. Due to the high level of the symmetry, we assume that all dynamic variables except the metric depend only on $t$ and $u^\mu = \delta_0^\mu$, so the constraint for the $\lambda_0$ is automatically satisfied.

For this geometry, $C_{\nu\lambda\sigma}^\mu = 0$; therefore, $T^{\mu\nu}[C^2] = 0$. Since $T_1^1 = T_2^2 = T_3^3$, we can only use $T^{00}$ and $T$. From the constraint for $\lambda_2$, we get:

$$\dot{X} = 0 \quad \Rightarrow \quad X = const, \tag{45}$$

The dot denotes the derivative with respect to $t$.

The equations of motion for the metric (44) and the action of matter in question are:

$$T^{00} = \varepsilon + \beta \lambda_1 \left\{ \Lambda \varphi^4 + \dot\varphi^2 + \varphi^2 \frac{\dot a^2 + k}{a^2} + 2\varphi\dot\varphi \frac{\dot a}{a} \right\} + \beta \dot\lambda_1 \varphi \left( \dot\varphi + \varphi \frac{\dot a}{a} \right) +$$
$$+ \lambda_1 \left( \gamma_1 \varphi n + \gamma_2 n^{\frac{4}{3}} \right) = 0, \quad (46)$$

$$\varphi\ddot\lambda_1 + \dot\lambda_1 \left( 3\varphi \frac{\dot a}{a} + 2\dot\varphi \right) + \lambda_1 \left( 2\ddot\varphi + 6\frac{\dot a}{a}\dot\varphi + 4\Lambda \varphi^3 - \frac{1}{3} \varphi R \right) + \lambda_1 \frac{\gamma_1}{\beta} n = -\frac{1}{\beta} \frac{\partial\varepsilon}{\partial\varphi}, \quad (47)$$

$$\Phi = \beta \varphi \left( \frac{1}{a^3} \frac{d}{dt} \left( a^3 \dot\varphi \right) - \frac{1}{6} \varphi R + \Lambda \varphi^3 \right) + \gamma_1 \varphi n + \gamma_2 n^{\frac{4}{3}} = \frac{1}{a^3} \frac{d}{dt} \left( a^3 n \right), \quad (48)$$

$$\frac{\partial\varepsilon}{\partial n} + \dot\lambda_1 + \lambda_1 \gamma_1 \varphi + \frac{4}{3} \lambda_1 \gamma_2 n^{\frac{1}{3}} = 0, \quad (49)$$

$$T = \varepsilon - 3p - \varphi \frac{\partial\varepsilon}{\partial\varphi} = 0, \quad (50)$$

where $R = -6 \frac{a\ddot a + \dot a^2 + k}{a^2}$ is a scalar curvature.

The system of equations under consideration is degenerate, since the equation of motion on $\varphi$ multiplied by $\dot\varphi + \varphi \frac{\dot a}{a}$ is obtained by differentiation with respect to $t$ equation for $T^{00}$ and using the rest. Thus, one of the equations can be eliminated, except for the case when $\dot\varphi + \varphi \frac{\dot a}{a} = 0$. An additional relation connecting the original equations is associated with the conservation of the energy–momentum tensor in quadratic gravity and, as a consequence, its special case—conformal gravity.

Let us consider the special case $\beta = 0$, in which the external scalar field is not dynamic, that is, the action does not contain derivatives $\varphi$. Moreover, from the equations, it follows that $\lambda_1 = const$, $\varepsilon = -\lambda_1 \left( \gamma_1 \varphi n + \gamma_2 , n^{\frac{4}{3}} \right)$, and the functions $a(t)$ and $\varphi(t)$ are arbitrary. It should be noted that from the birth law in this case it follows that at $n = 0$, $\dot n$ is also equal to zero; that is, in order for the birth of particles from the vacuum to begin, there must be a contribution from the term at $\beta$ or geometry for which $C^2 \neq 0$.

As stated above, the conformal invariance of the action and, as a consequence, the equations of motion allow one to arbitrarily choose the gauge.

Let us suppose that we found somehow the specific solution for the set of dynamical variables $\{\hat a, \hat n, \hat\varphi\}$. Then, the general solution is $\{a, n, \varphi\}$, where $a = \hat a \Omega$, $\hat n = n \Omega^3$, $\hat\varphi = \varphi \Omega$ with arbitrary smooth function $\Omega(t)$. One can use such a freedom to choose the most appropriate gauge.

The free choice of gauge forces us to think about which of them is physical, that is, which is most consistent with the accumulated observational data. In the fourth section, we considered Equation (21), which follows from the conformal invariance of the action of gravity, for two special cases—dust and radiation. For dust, we found that the effective mass of particles, a factor of $n$, depends on the external scalar field, and in the general case is not constant. In this regard, we can assume that the gauge $\varphi = const$ is physical.

Below we will write the set of equations for two different gauges: $\varphi = \varphi_0 = const$ and $\hat a = 1$. The latter does not mean at all that the "real" Universe is static. It is chosen because in such a case the set of equations looks simplest. However, in our opinion, the gauge $\varphi = \varphi_0$ may be considered physical since the mass of the dust particles become constant.

Transition from the "comfortable" gauge $\hat a = 1$ to the physical gauge $\varphi = \varphi_0$ can be easily achieved in the following way. Since $a \varphi = \hat a \hat\varphi$, we have $a \varphi = \hat\varphi(\eta)$ and $a(\eta)^2 d\eta^2 = dt^2$, where $\eta$ is the conformal time, and t is the cosmological time.

Let us consider the gauge $\varphi = \varphi_0$:

$$0 = T^{00} = \varepsilon + \beta \lambda_1 \left\{ \Lambda \varphi_0^4 + \varphi_0^2 \frac{\dot a^2 + k}{a^2} \right\} + \beta \dot\lambda_1 \varphi_0^2 \frac{\dot a}{a} + \lambda_1 \left( \gamma_1 \varphi_0 n + \gamma_2 n^{\frac{4}{3}} \right), \quad (51)$$

$$\varphi_0 \ddot{\lambda}_1 + 3\dot{\lambda}_1 \varphi_0 \frac{\dot{a}}{a} + \lambda_1 \left(4\Lambda \varphi_0^3 - \frac{1}{3}\varphi_0 R\right) + \lambda_1 \frac{\gamma_1}{\beta} n = -\frac{1}{\beta}\frac{\partial \varepsilon}{\partial \varphi}, \tag{52}$$

$$\beta \varphi_0 \left(-\frac{1}{6}\varphi_0 R + \Lambda \varphi_0^3\right) + \gamma_1 \varphi_0 n + \gamma_2 n^{\frac{4}{3}} = \frac{1}{a^3}\frac{d}{dt}\left(a^3 n\right), \tag{53}$$

$$\frac{\partial \varepsilon}{\partial n} + \dot{\lambda}_1 + \lambda_1 \gamma_1 \varphi_0 + \frac{4}{3}\lambda_1 \gamma_2 n^{\frac{1}{3}} = 0, \tag{54}$$

$$4\varepsilon - 3n \frac{\partial \varepsilon}{\partial n} - \varphi_0 \frac{\partial \varepsilon}{\partial \varphi} = 0. \tag{55}$$

For $k = 0$, there is a particular solution with $n = n_0 = const$:

$$\frac{\dot{a}}{a} = \frac{3n_0}{\beta \varphi_0^2} - \gamma_1 \varphi_0 - \frac{4}{3}\gamma_2 n_0^{\frac{1}{3}}, \tag{56}$$

$$\beta \varphi_0^2 \left(\Lambda \varphi_0^2 + 2\left(\frac{3n_0}{\beta \varphi_0^2} - \gamma_1 \varphi_0 - \frac{4}{3}\gamma_2 n_0^{\frac{1}{3}}\right)^2\right) + \gamma_1 \varphi_0 n_0 + \gamma_2 n_0^{\frac{4}{3}} =$$

$$= 3n_0 \left(\frac{3n_0}{\beta \varphi_0^2} - \gamma_1 \varphi_0 - \frac{4}{3}\gamma_2 n_0^{\frac{1}{3}}\right), \tag{57}$$

$$\varepsilon(n_0, \varphi_0) = \beta \varphi_0^2 \left(\frac{3n_0}{\beta \varphi_0^2} - \gamma_1 \varphi_0 - \frac{4}{3}\gamma_2 n_0^{\frac{1}{3}}\right) \frac{\partial \varepsilon}{\partial n}(n_0, \varphi_0). \tag{58}$$

Here we can draw an analogy with the solution with $k = 0$ obtained in the work of [23]. Transition to conformal time:

$$d\eta^2 = \frac{dt^2}{a(t)^2}, \quad ds^2 = a^2(\eta)\left\{d\eta^2 - \frac{dr^2}{1 - kr^2} - r^2 d\Omega^2\right\}, \tag{59}$$

This allows us to choose a gauge $a(\eta) = 1$ in which $\eta = t$. The equations of motion for this gauge are as follows:

$$T^{\eta\eta} = \beta\left(\varphi \dot{\lambda}_1 \dot{\varphi} + \lambda_1 \dot{\varphi}^2 + \lambda_1 \varphi^2 \left(k + \Lambda \varphi^2\right)\right) + \varepsilon + \lambda_1 \left(\gamma_1 \varphi n + \gamma_2 n^{\frac{4}{3}}\right) = 0, \tag{60}$$

$$\frac{1}{\beta}\frac{\partial \varepsilon}{\partial \varphi} + 2\lambda_1 \ddot{\varphi} + 2\dot{\lambda}_1 \dot{\varphi} + \varphi \ddot{\lambda}_1 + \lambda_1 \varphi \left(2k + 4\Lambda \varphi^2\right) + \frac{\gamma_1}{\beta}\lambda_1 n = 0, \tag{61}$$

$$\frac{\partial \varepsilon}{\partial n} + \dot{\lambda}_1 + \lambda_1 \gamma_1 \varphi + \frac{4}{3}\lambda_1 \gamma_2 n^{\frac{1}{3}} = 0, \tag{62}$$

$$\dot{n} = \beta\left(\varphi \ddot{\varphi} + k \varphi^2 + \Lambda \varphi^4\right) + \gamma_1 \varphi n + \gamma_2 n^{\frac{4}{3}}, \tag{63}$$

$$4\varepsilon - 3n \frac{\partial \varepsilon}{\partial n} - \varphi \frac{\partial \varepsilon}{\partial \varphi} = 0. \tag{64}$$

Let us consider the special case $\lambda_1 = const$:

$$\varepsilon = -\lambda_1 \left(\Phi - \beta \varphi \ddot{\varphi} + \beta \dot{\varphi}^2\right), \tag{65}$$

$$\frac{\partial \varepsilon}{\partial n} = -\lambda_1 \frac{\partial \Phi_1}{\partial n}, \tag{66}$$

$$\dot{n} = \Phi, \tag{67}$$

$$4\varepsilon - 3n \frac{\partial \varepsilon}{\partial n} - \varphi \frac{\partial \varepsilon}{\partial \varphi} = 0. \tag{68}$$

Only four equations are used here because, as noted earlier, not all of the original equations are independent. From the condition $T = 0$ in this case it follows:

$$\varepsilon = -\lambda_1 \left( \gamma_1 \, \varphi \, n + \gamma_2 \, n^{\frac{4}{3}} \right) + C \varphi^4, \tag{69}$$

where $C$ is some constant. Wherein field $\varphi$ satisfies the equation:

$$\dot{\varphi}^2 = -k \, \varphi^2 - \left( \frac{C}{\beta \lambda_1} + \Lambda \right) \varphi^4. \tag{70}$$

For $k = 0$:

$$\varphi = \frac{\sigma}{\sqrt{-\left( \frac{C}{\beta \lambda_1} + \Lambda \right)}} \frac{1}{\eta + C_0}, \tag{71}$$

where $\sigma = \pm 1$ is a sign of $\dot{\varphi}$; for $k = \pm 1$ we have, respectively:

$$\sigma \, arctg \left( \frac{1}{\sqrt{-\left( \frac{C}{\beta \lambda_1} + \Lambda \right) \varphi^2 - 1}} \right) = \eta + C_0, \quad k = 1, \tag{72}$$

$$\sigma \, arcth \left( \frac{1}{\sqrt{-\left( \frac{C}{\beta \lambda_1} + \Lambda \right) \varphi^2 + 1}} \right) = \eta + C_0, \quad k = -1, \tag{73}$$

where $C_0$ is a constant depending on the initial conditions.

The scale factor $a(\eta)$ changes as follows under the conformal transformation $a = \Omega \, \hat{a}$; therefore, when going to the gauge $\hat{a} = 1$, $a = \Omega$. If initially $\varphi = \varphi_0 = const$, then $\hat{\varphi} = \varphi_0 \Omega = \varphi_0 a$; that is, the scalar field calculated in the gauge $\hat{a} = 1$, proportional to the scale factor in the gauge $\varphi = \varphi_0 = const$. In particular, the result obtained above for $\lambda_1 = const$ is consistent with that calculated earlier, since, when moving to the gauge $\varphi = \varphi_0 = const$, the scalar curvature remains constant:

$$R = -6 \frac{a'' + a \, k}{a^3} = 12 \left( \Lambda + \frac{C}{\beta \lambda_1} \right) \varphi_0^2, \tag{74}$$

where the prime denotes the derivative with respect to $\eta$ and $a(\eta) = \frac{1}{\varphi_0} \hat{\varphi}(\eta)$ with function $\hat{\varphi}(\eta)$ defined by the Equations (71), (72) or (73) depending on k.

Let us consider the transition to the variable t from $\eta$ for the case $k = 0$:

$$a = \frac{1}{\varphi_0} \frac{\sigma}{\sqrt{-\left( \frac{C}{\beta \lambda_1} + \Lambda \right)}} \frac{1}{\eta + C_0} \propto exp \left\{ \sigma \, \varphi_0 \sqrt{-\left( \frac{C}{\beta \lambda_1} + \Lambda \right)} \, t \right\}, \tag{75}$$

where we have chosen the minus sign in the relation: $dt = -a(\eta) d\eta$. Thus, if $\sigma \varphi_0 > 0$, we obtain exponential growth for the scale factor $a(t)$. Moreover, from Equation (70), it follows that the same is true for $k = \pm 1$ when $\hat{\varphi} \to \infty$, which is equivalent to $t \to \infty$. This is due to the fact that with $\lambda_1 = const$ in the gauge $\varphi = \varphi_0$ for the homogeneous and isotropic space-time with the metric (44), our model actually reduces to general relativity with the cosmological constant.

## 7. Discussion

The conformal invariance of the term in the action of matter, from which the particle creation law is obtained, leads to restrictions on the invariants of external fields responsible for the creation processes on which the function $\Phi$ depends. In the absence of classical

external fields, when the only source of particle creation is gravity, the square of the Weyl tensor is the most basic option. Due to this fact, conformal invariance of the gravity action leads to a case in which the total action is equivalent to the matter action up to redefining Lagrange multiplier $\lambda_1$.

When an external scalar field is introduced into the creation law, the following combination is chosen: $\varphi \Box \varphi - \frac{1}{6} \varphi^2 R + \Lambda \varphi^4$, since it yields a nontrivial equation of motion and is conformally invariant when multiplied by $\sqrt{|g|}$.

In addition to the above, contributions to the creation law proportional to the particle number density are also possible. In cosmology, the $\gamma_1$ term can be interpreted as a dark matter. It is not real matter, but the "memory" of the process of the particle production. The conditions for its existence are $n \neq 0 \, (> 0)$. Thus, the real particles should already be produced. The dark matter will exist even after the particle creation stops. The $\gamma_2$ term becomes the hot universe, even without real photons and real temperature. Both of them are just images, but they are gravitating.

This interpretation is possible due to the fact that, in the hydrodynamic part of the energy–momentum tensor, the terms with $\gamma_1$ and $\gamma_2$ are not associated with any matter, but indicate the influence on gravity of the particle creation process itself, which can be used to explain the "missing" mass in the Universe. Moreover, their contribution is in many ways similar to the contribution from dust and radiation, which unites our model with the one presented in the article [12], where the matter creation also makes a contribution to the energy–momentum tensor similar to an ideal fluid.

As mentioned in the introduction, the phenomenological description of particle creation in cosmology is best suited to the early Universe. However, the solution obtained in our model for $\lambda_1 = const$ (75) shows that it is also applicable for the present phase of the evolution of the Universe.

In the absence of a scalar field, the matter under consideration within this model, when the action of gravity is conformally invariant, can only be radiation. For cosmological solutions, by which we mean homogeneous and isotropic geometry, without a scalar field the creation of particles cannot begin from the vacuum. On the other hand, if $n \neq 0$ or $\varphi \neq 0$, then, unlike the models discussed in the articles [19–21], particle production is possible even in homogeneous and isotropic geometry, where the square of the Weyl tensor is zero.

From the conformal invariance of the gravity action for the model with an external scalar field it follows that, for dust, the energy density is proportional to the scalar field, while for radiation, it does not depend on the scalar field. Therefore, the gauge $\varphi = const$ seems to be the most consistent with the observational data.

**Author Contributions:** The authors (V.B. and I.I.) contributed equally to this work. All authors have read and agreed to the published version of the manuscript.

**Funding:** This research received no external funding.

**Data Availability Statement:** No new data were created or analyzed in this study. Data sharing is not applicable to this article.

**Conflicts of Interest:** The authors declare no conflicts of interest.

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
