# Peer review of "Conformally Invariant Gravity and Gravitating Mirages"

_universe, doi:10.3390/universe10030147_

Round 1

Reviewer 1 Report

Comments and Suggestions for Authors

The paper titled "Conformally invariant gravity and gravitating mirages" by Victor Berezin and Inna Ivanova, explores the phenomenological description of particle creation in strong external fields through the action of an ideal fluid in Euler variables. It demonstrates how conformal invariance imposes strict restrictions on possible sources, suggesting that certain combinations in the creation law can be interpreted as dark matter. 

Sections 1-5 adopt a standard approach to deriving equations of motion and constraints for the action given by Eq. (17).

The challenges within the paper arise in Sec 6:

A static model of the Universe is suggested (a=1) without providing physical justification or any rationale. There is no attempt to compare the results of this paper with the standard cosmological model.

Section 7 introduces dark matter without adequate explanation or references.

No clear links to the paper's text are provided for these results either.

For consideration of publication, sections 6 and 7 require substantial revision.

Points to Address in Revised Paper:

1. Provide a clear physical or theoretical justification for the static model of the Universe proposed in Sec 6.

2. Compare and contrast the results with the standard cosmological model to highlight the contributions and deviations of this paper.

3. Elaborate on the discussion of dark matter in Sec 7, including relevant explanations and references to existing literature.

4. Ensure all results are clearly linked to the text of the paper, facilitating reader comprehension and verification of claims.

Author Response

We wish to thank you for the time and effort dedicated to providing valuable feedback on our manuscript. Your suggestions and comments were useful and we hope that they will help improve it.

  1. If we found somehow the specific solution for the set of dynamical variables {a,n, φ}. Then the general solution is: {a,n, φ}, where a=a Omega, n= n Omega^3, φ = φ Omega with arbitrary smooth function Omega(t). One can use such a freedom to choose the most appropriate gauge. In the section 6 we consider the set of equations for two different gauges: φ = φ_0=const  and  a=1. The latter does not mean at all that the "real" Universe is static. It is chosen because in such a case the set of equations looks simplest. But, in our opinion, the gauge φ = φ_0 may be considered physical one since the mass of the dust particles become constant. Transition from the "comfortable" gauge a=1 to the physical gauge φ = φ_0 can be easily done in the following way. Since a φ= a φ, we have a φ= φ and a(eta)^2 d eta^2=dt^2, where eta is the conformal time and t is the cosmological time.
  2. When applying the model under consideration for cosmology, it can be assumed that it is most relevant for those phases of the evolution of the Universe when there was a rapid birth of particles. For example, immediately after the supposed birth of the Universe ``nothing'' or at the end of inflation during the reheating. Moreover, if the universe was born anisotropic, then, as shown in the articles (Zel’dovich, Ya. B.,Starobinsky, A. A. Sov. Phys. JETP 1972; Zel’dovich, Ya. B. JETP Lett. 1970; Lukash, V. N., Starobinsky A. A. Sov. Phys. JETP 1974), it was the birth of particles that led to its isotropization. However, the solution obtained in our model for lambda_1= const where we obtain exponential growth for the scale factor a(t) shows that it is also applicable for the present phase of the evolution of the Universe. This is due to the fact that with lambda_1=const in the gauge φ = φ_0 for the homogeneous and isotropic space-time, our model actually reduces to general relativity with the cosmological constant.
  3. The interpretation as dark matter is possible due to the fact that in the hydrodynamic part of the energy-momentum tensor the terms with gamma_1 and gamma_2 are not associated with any matter, but indicate the influence on gravity of the particle creation process itself, which can be used to explain the "missing" mass in the Universe. Moreover, their contribution is in many ways similar to the contribution from dust and radiation, which unites our model with the one presented in the article (J.Farnes 2018), where the matter creation also makes a contribution similar to an ideal fluid to the energy-momentum tensor.
  4. Please, see the revised version of the manuscript.

Reviewer 2 Report

Comments and Suggestions for Authors

In the manuscript the authors investigate matter creation by adopting a perspective  based on conformal invariance. The constraints on the matter production are included via the use of Lagrange multipliers. Cosmological applications are also presented. The manuscript may be publishable in Universe if the authors would fully consider the following points:

1. What is the physical interpretation of the variable X in the action (8)?

2. What is the physical origin of particle creation? This is essentially a quantum process, however, the authors use a classical description. Thus, there seems to be a contradiction between the real physics, and the adopted formalism, which is at most a phenomenological description of a complex process. The  authors must fully clarify this issue, and indicate the range of validity of the considered approach.   

3. Can the "creation law" (16) be interpreted quantum mechanically?

4. The cosmological applications must be significantly improved. First of all, it is not clear in what phase of the evolution of the Universe particle creation may be important (inflation, early Universe, late Universe). Secondly, the authors must present some quantitative details on the variation of the scale factor, of the scalar field, and of the matter energy density, at least. A comprehensive cosmological picture in the presence of matter creation must be presented.

5. Does matter creation have a maximum, and, if yes, when does it occur?

6. The Introduction and Discussion Sections of the manuscript, as well as the references must be significantly extended. A discussion of other approaches to matter creation must also be provided.  

Comments on the Quality of English Language

Quality of English language acceptable.

Author Response

We wish to thank you for the time and effort dedicated to providing valuable feedback on our manuscript. Your suggestions and comments were useful and we hope that they will help improve it.

  1. Dynamical variable X(x) is introduced in order to avoid the identically zero vorticity of particle flow. From the constraint with the Lagrangian multiplier lambda_2 it follows that X(x) is constant along the trajectories and therefore the choice of this function defines the labelling of the trajectories.
  2. Particle creation is a quantum process but classical description is possible when the external fields are strong enough and the separation between just created particles becomes of order of their Compton length, and we can safely approximate them by some condensed matter. For example F. Hoyle (1962) used classical creation field in order to introduce the idea of continuous creation of matter. The thermodynamic approach to particle production at the expense of a gravitational field has been studied in (I. Prigogine et al 1988). Recently J. Farnes (2018) applied Hoyle's creation tensor and the concept of negative mass to propose a single negative-mass fluid explanation of dark matter and dark energy. In the phenomenological approach to the particle creation the nonlocal processes become, formally, the local ones. The same concerns also the trace anomalies and for example in the article (T.D. Netto et al. 2016) it is shown that the non-local terms in the effective action become insignificant under certain conditions. In this case, the use of anomaly induced effective action can be considered as an example of a phenomenological description of particle production.
  3. In the absence of classical external fields, the birth of particles is due to the vacuum polarization caused by gravity, so Phi is a function of geometric invariants. In Riemannian geometry in the four-dimensional case, the square of the Weyl tensor is the only possible choice if we restrict ourselves to invariants , which are at most quadratic in the curvature tensor. The same result was obtained in (Zel’dovich Ya. B., Starobinskii A. A, 1977) for the particle creation by the vacuum fluctuations of the massless scalar field on the background of the homogeneous and slightly anisotropic cosmological spacetime. In this regard, it can be assumed that the creation law describes the relationship between the vacuum average values of the corresponding quantities.
  4. When applying the model under consideration for cosmology, it can be assumed that it is most relevant for those phases of the evolution of the Universe when there was a rapid birth of particles. For example, immediately after the supposed birth of the Universe ``nothing'' or at the end of inflation during the reheating. Moreover, if the universe was born anisotropic, then, as shown in the articles (Zel’dovich, Ya. B.,Starobinsky, A. A. Sov. Phys. JETP 1972; Zel’dovich, Ya. B. JETP Lett. 1970; Lukash, V. N., Starobinsky A. A. Sov. Phys. JETP 1974), it was the birth of particles that led to its isotropization. However, the solution obtained in our model for lambda_1= const where we obtain exponential growth for the scale factor a(t) shows that it is also applicable for the present phase of the evolution of the Universe.
  5. We can investigate the case when the left-hand side of the creation law is zero. In the gauge φ = φ_0, it is reduced to the condition n = n_0, from which a differential equation for φ follows that can be solved analitically. However, the search for the maximum of matter creation, that is, the extremum of the left-hand side of the creation law, is unlikely to be reduced to a problem that can be solved analytically, and numerical modeling will probably require more time.
  6. Please, see the revised version of the manuscript.

Round 2

Reviewer 1 Report

Comments and Suggestions for Authors

The revised paper claims that conformally static solutions are fundamentally different from non-static solutions, where the conformal factor, Ω, varies with time. This distinction appears to be a misunderstanding or misrepresentation in the context of conformally invariant theories.

Conformal transformation indicates that physical laws remain unchanged under metric rescaling.

A key point of conformal invariance is that any solution to the equations, and its conformally related counterparts—obtained by applying a conformal transformation with Ω depending on time, space, or both—are physically equivalent. Thus, the difference between "static" and "non-static" solutions is merely a choice of coordinates or frames, not a substantial physical variation. Although the authors acknowledge this equivalence (Lines 226-229), their conclusions suggest otherwise. The theory does not distinguish between these solutions; it considers them equivalent as they depict the same physical situation through different conformal frames.

The assertion (Lines 236-242) that conformally static solutions are distinct from non-static solutions (where Ω is time-dependent) overlooks the essence of conformal invariance. The time-dependence of Ω does not change the solution's physical essence. Instead, it influences the representation or interpretation of the solution within a particular coordinate system or frame.

Author Response

Thank you for your comment, we will try to answer it. If an action has some symmetry, this does not necessarily mean that the solutions to the equations of motion are physically equivalent if they differ only in the transformation of the corresponding symmetry. The only thing that can be stated is that if there is a specific solution to the equations of motion, then the symmetry transformation allows us to obtain another solution.

Reviewer 2 Report

Comments and Suggestions for Authors

The authors have improved their work, and hence I think the present version is suitable for publication in Universe.

Comments on the Quality of English Language

Quality of English language is good.

Author Response

Thank you very much for your contribution.